# Potential Fertilization Capacity of Two Grapevine Varieties: Effects on Agricultural Production in Designation of Origin Areas in the Northwestern Iberian Peninsula

**J. Antonio Cortiñas [1],*** , **María Fernández-González [1,2]** , **Estefanía González-Fernández [1]** ,
**Rosa A. Vázquez-Ruiz [3]** , **F. Javier Rodríguez-Rajo [1]** and **María Jesús Aira [4]**

1 CITACA, Agri-Food Research and Transfer Cluster, Faculty of Sciences, University of Vigo,
  32002 Ourense, Spain; mfgonzalez@uvigo.es (M.F.-G.); estegonzalez@uvigo.es (E.G.-F.);
  javirajo@uvigo.es (F.J.R.-R.)
2 Earth Sciences Institute (ICT), Pole of the Faculty of Sciences University of Porto, 4169-007 Porto, Portugal
3 Department of Botany, Higher Polytechnic School of Engineering, University of Santiago de Compostela,
  27002 Lugo, Spain; rosana.vazquez@usc.es
4 Department of Botany, Pharmacy Faculty, University of Santiago de Compostela,
  15782 Santiago de Compostela, Spain; mariajesus.aira@usc.es
* Correspondence: jcortinas@uvigo.gal

**Abstract:** In the present study, we analyzed the main parameters related with the potential fertilization ability of two grapevine varieties, Godello and Mencía, during the years 2017 and 2018. The research was carried out in two vineyards of the Galician winegrowing Designation of Origin areas of Ribeiro and Ribeira Sacra. Ten vines of each variety were selected for bunch and flower counting, pollen calculations, pollen viability studies by means of aceto-carmine (AC) stain and 2, 3, 5-triphenyl tetrazolium chloride (TTC) methods, and the determination of their germination rate. In all vineyards the 50% fruitset was reached, except for Godello in Cenlle during 2017. The mean coulure value was higher for Godello (40.5%) than for Mencía (31%). Analyzing the pollen production per plant and airborne pollen levels, we observed important discordances between them, which can be due to the influence of weather conditions and be related with self-pollination processes. We found important differences on pollen viability depending on the applied method and variety, with higher values for the AC method than the TTC for both varieties in all study plots, and higher values for Mencía variety than Godello. Regarding germination rates, we observed a marked reduction in 2017 with respect to 2018, in all study sites and for both varieties. The analyzed parameters were useful to explain the different productive abilities of Godello and Mencía varieties in the two studied bioclimatic regions of Ribeiro and Ribeira Sacra.

**Keywords:** pollen fertility; Godello; Mencía; reproductive performance; harvest production

## 1. Introduction

Vine growing is closely associated to climatology, which means that climate change effects have become of great concern, promoting new research lines focused on the adaptability of the different grapevine varieties to the new scenarios [1,2]. As temperature increases, rainfall patterns will change and wine production will, therefore, need to adapt at short-term future [3]. Some authors state that it will be necessary for territorial relocations of vineyards or the introduction of drought-resistant varieties [4].

Nevertheless, crop yield not only depends on climatology. The *terroir* characteristics, crop management practices (e.g., pruning, tillage, planting system, or irrigation), and the cultivated grapevine variety [5–7]



are important factors to considered both independently and as a whole [8]. The genetic characteristics of the different grapevine varieties determine their resistance to frost and cryptogamic disease, such as powdery mildew, grey mold, or downy mildew, which can cause serious economic losses [9,10]. The epidemiological characteristics of the fungi responsible for these pathologies depend, among other factors, on the airborne spore concentrations [11]. Therefore, to know the host resistance is useful for the long-term management of the disease [12,13]. Many works on grapevine phytopathology have been developed during the last years, providing valuable information for crop protection in order to ensure a successful vintage [14–16]. However, fertilization-related aspects are also determinants for crop production. The different grapevine varieties use distinct mechanism to ensure optimal ovary fertilization and fruit development. Some of them are self-incompatible (such as Burdur dimriti variety with hermaphrodite flowers, and Buzgulu variety with morphological hermaphrodite-physiological female flowers, both grown in Isparta, Turkey [17]), pollinated by wind or insects, although the low nectar production and the absence of special flower characters limits its attraction for insects [18]. The starch accumulation in pollen grains was demonstrated to be related to semi-fertile varieties [19]. Pollen polymorphism, which is a widespread phenomenon among higher plants [20,21], could be also related to the irregular productivity of some grapevine cultivars [19–22].

These reproduction functioning variations among the different grapevine varieties are reflected, as well, on reproductive performance variables, such as the number of flowers and berries per inflorescence/bunch or the production of pollen grains by the plant. A recent study characterized the reproductive performance of 120 grapevine varieties, the largest comparative study done in this field, in which a great diversity was found for most variables, including fruitset and the number of flowers [23]. They obtained a clustering of similar varieties, in which groups were related with the variety use (table, wine, or both) and its geographical origin. Varieties included in these classes showed a non-random distribution regarding genetic structure based on molecular markers, which suggests the importance of genetic regulation on reproductive performance variables.

Pollen viability is other relevant factor related to grapevine yield. Despite pollen usually being produced in a large excess per individual flower [18], which ensures pollination and fruitset even at low pollen viability rates, pollen sterility might appear with a certain level of ovule sterility, which acts as a limiting factor in fruitset and grape production since each flower only produces four ovules [24]. Nevertheless, low pollen viability under certain conditions may limit grapevine yield, leading to important economic losses [25]. Baby et al. [26] found that variations in reproductive performance of the three studied grapevine varieties were related to differences in pollen viability and amine concentrations in the reproductive organs, since they found that the poor reproductive performance of the Cabernet Sauvignon and Merlot varieties compared with that of Shiraz variety was correlated with poor pollen viability. This seems to be related with amine concentrations, given that a significantly higher concentration of diaminopropane was found in the two varieties with poor reproductive performance, and correlated with a higher proportion of underdeveloped berries.

Pollen grain development is sensitive to environmental conditions, both abiotic (such as heat, cold, or drought events) and biotic factors [27]. In the case of *Vitis vinifera* pollen, its development is highly sensitive to temperature stress [28]. Although meiosis occurs normally during heat events, this results in a significant reduction in pollen viability attributable to highly disrupted pollen grains, probably due to cell wall fragility. The effect of heat events studied by Pereira et al. [28], with exposure of plants to 42 °C for 4 h, are included in the temperature range in which grapevine usually grows, and this induces gametophyte malformations. In vitro germination assays showed that heat stress affected, as well, the germination capacity, being reduced by approximately half in comparison with control plants not exposed to high temperatures [28].

Considering that both pollen quantity and quality are essential factors to ensure a good grape production, several studies have been conducted on pollen production, viability, germination, and growth of the pollen tube [18–32]. Aerobiological studies are also valuable since atmospheric pollen concentrations during the grapevine flowering are a measurement linked to the number of flowers

and fruits [33], depending on environmental variables [34]. Based on these studies, the objective of the present work was to analyze the main influential parameters on the yield of two grapevine varieties (Godello and Mencía), mainly those that have an impact on the fertilization process.

## 2. Materials and Methods

### 2.1. Situation of the Study Area and Climatic Characteristics

The study was carried out in three vineyards located in Cenlle (Ribeiro region) and Doade and Souto (Ribeira Sacra region, Spain) during 2017 and 2018 (Figure 1). The Ribeiro region covers an area of 2600 hectares of vineyards. The study plot is situated in the highest part of the mountain and some distance from the Avia river (altitude: 117 m above mean sea level; 42°18′55.7″ N, 8°6′2.54″ W). The soils are medium depth, between 70 and 100 cm, of granitic origin, and stony, which favors the drainage and insolation of bunches. The Ribeira Sacra vineyards cover about 2500 hectares. The peculiar orography of this territory forces the wine cultivation on very steep slopes with incline up to 80%. The Doade vineyard is placed on terraces of the upper part of the hillside (551 m above mean sea level; 42°24′31.9″ N, 7°28′33.7″ W), while the Souto vineyard is located in the low area, next to the Sil river (438 m above mean sea level; 42°24′27.67″ N, 7° 28′20.06″ W). In both cases, the soils are shallow and have low fertility, which limits the vine shoots' growth [35].

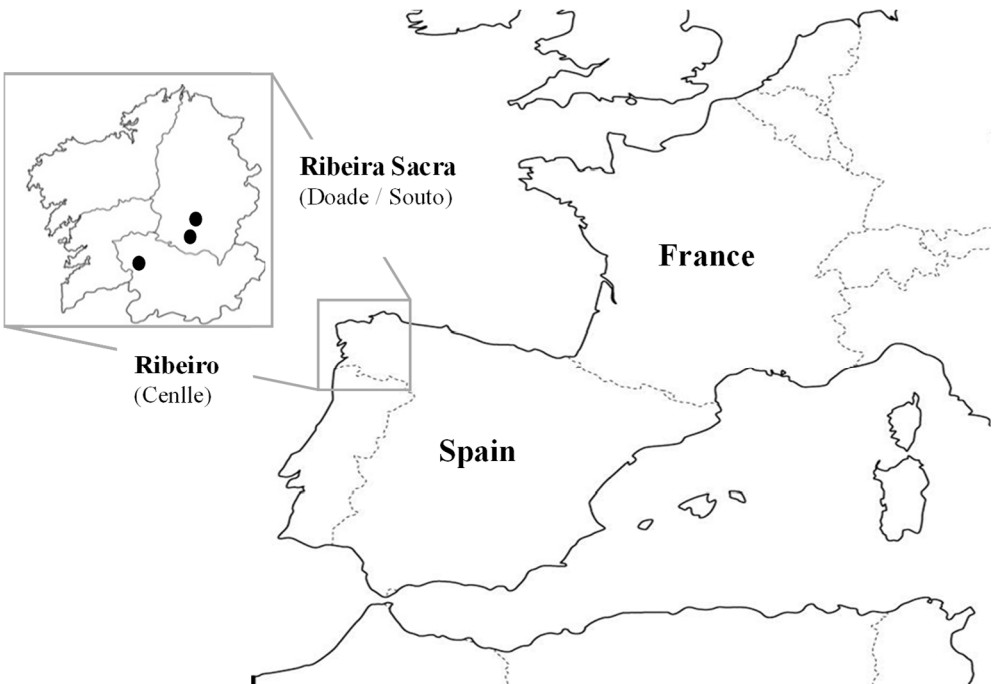

**Figure 1.** Location of the Ribeiro (Cenlle vineyard) and Ribeira Sacra (Doade and Souto vineyards) winegrowing Designation of Origin areas.

The selected grapevine varieties were Godello and Mencía. Godello is a preferential cultivar in both regions, which has high vigor and erect bearing, with early sprouting, bud development, and maturation. The Mencía cultivar is mainly grown in Ribeira Sacra. It has medium vigor, with early sprouting and semi-late ripening.

The Godello variety has small- to medium-sized bunches, with a conical shape and fairly homogeneously sized berries of medium to high compactness, with a medium-sized peduncle and little lignification at its base. The leaf is medium to large, pentagonal in shape, and has five lobes [36].

The Mencía variety is characterized by its medium-sized, compact, cone-shaped bunches. The stem of the bunch is quite long. The berries are spherical, with a medium-size, ellipsoidal shape, regular in its

cross-section, thick skin, and very striking, bluish-black color. The leaf has a medium and pentagonal blade size, with five lobes of straight-convex teeth [36].

The flower of the vine plant has no visible petals. Instead, the petals fuse into a green structure called calyptra. The calyptra includes the reproductive organs and other tissues within the flower. A flower consists of a single pistil (female organ) and five stamens, each with an anther at the tip (male organ). The pistil is approximately conical in shape, with the base much larger than the top, and the end (called the stigma) is slightly flared. The base of the pistil is the ovary, and consists of two internal compartments, each containing two eggs. The anthers produce a large number of pollen grains. Most commercial vine varieties have hermaphrodite flowers, i.e., the two male and female components [36].

From the phytopathological point of view, both varieties are sensitive to *Erysiphe necator* (Schwein.) and *Plasmopara viticola* (Berk. & Curtis) Berl. & de Toni, and to a lesser extent to *Botrytis cinerea* Pers.: Fr. (teleomorph: *Botryotinia fuckeliana* (de Bary) Whetzel) [36]. We considered the meteorological data of maximum (Tmax, °C), minimum (Tmin, °C), and average temperature (Tmean, °C), relative humidity (mean RH, %), and sunshine hours and rain (L/m$^2$) provided by two stations located very close to the vineyards (Leiro and Xábrega) for climatic characterization (http://www.meteogalicia.gal).

To classify both regions regarding temperature variations and the amplitude of temperature annual oscillation, we calculated the continentality index (Ic) [37]. The Ic index was calculated as:

$$Ic = Tmax - Tmin \tag{1}$$

where Tmax is the average monthly mean temperature of the hottest month of the year, in °C, and Tmin is the average monthly mean temperature of the coldest month of the year, in °C.

*2.2. Field Experimental Tasks*

We selected 10 vines of the studied varieties on each vineyard. The total number of clusters was counted during flowering, and the number of flowers and berries (both set and ripe) were counted in a control cluster marked on each vine. The number of flowers per vine (flowers/vine) was estimated considering the floral value and the number of clusters. We calculated the average value of the analyzed variables eliminating the highest and lowest values to homogenize the data [38]. For calculation of the fruitset and coulure indices, we used the following expressions:

$$\text{Fruitset \%} = \frac{\text{berry number per bunch}}{\text{number of flowers per inflorescence}} \times 100 \tag{2}$$

$$\text{Coulure \%} = \frac{(\text{number of flowers per inflorescence}) - (\text{number of berries per bunch})}{\text{number of flowers per inflorescence}} \times 100 \tag{3}$$

Fruitset and coulure indices are complementary values in this work (referred to 100%), since we did not find seedless berries or live green ovaries (LGO) in the studied clusters, counting only seeded berries. In addition, we collected flower samples of all analyzed vines to separate the anthers for palynological studies carried out in laboratory.

Finally, to monitor the airborne *V. vinifera* pollen concentrations in the atmosphere of the studied vineyards, Hirst-type volumetric samplers [39], model Lanzoni VPPS 2010 (Lanzoni s.r.l., Bologna, Italy), were used. For sample processing, we followed the Spanish Aerobiological Network protocol [40] of four longitudinal transects along the microscope slides at 400× magnification using a light optical microscope (Nikon Instruments Europe B.V., Amsterdam, Netherlands). Average daily concentrations of pollen/m$^3$ of air were summed over each season to obtain the annual number of airborne pollen grains [41].

Pollen grains of the common cultivated grapevine *V. vinifera* L. subsp. *vinifera* are normally 3-zonocolporate, spheroidal to prolate, with very long, narrow, slit-like ectoaperture (colpus), and endoaperture of circular pore [21].

### 2.3. In Vitro Assays and Laboratory Analysis

The *V. vinifera* pollen production, viability, and germination ability were measured for samples of the both grapevine varieties considered from the Cenlle, Doade, and Souto vineyards. To calculate the number of pollen grains per vine (pollen/vine) we isolated two anthers of two flowers from 10 different plants. A Neubauer chamber (Brand™ 717820, Fisher Scientific SL, Madrid, Spain) was used for pollen counting [42,43]. The resulting value was divided by 2 due to the number of anthers used [43,44]. It was then multiplied by 5, corresponding to the number of anthers per flower [45], and by the total number of flowers on each vine.

Pollen viability and germination rates were determined on 10 samples for each variety, method, vineyard, and year. Two different colorimetric methods were applied for pollen viability determination. The used stains for the study of pollen viability were aceto-carmine (AC, PanReac Química SLU, Barcelona, Spain) and 2, 3, 5-triphenyltetrazolium chloride (TTC, Merck Millipore, Madrid, Spain), identifying the damaged cells as non-viable pollen (Figure 2). The TTC is a redox indicator especially indicative of cellular respiration that requires previous incubation in a humid chamber and in the dark during 24–48 h at 25 °C. The acetic carmine has traditionally been used for staining chromosomes and does not require incubation before observation [46]. In both cases, the colorimetric reaction was observed under a light microscope ×100 (Olympus Iberia S.A.U., Barcelona, Spain) at least on 100 pollen grains per preparation. The results were expressed as percentages regarding the ratio of the number of viable pollen grains to the total number of grains [47].

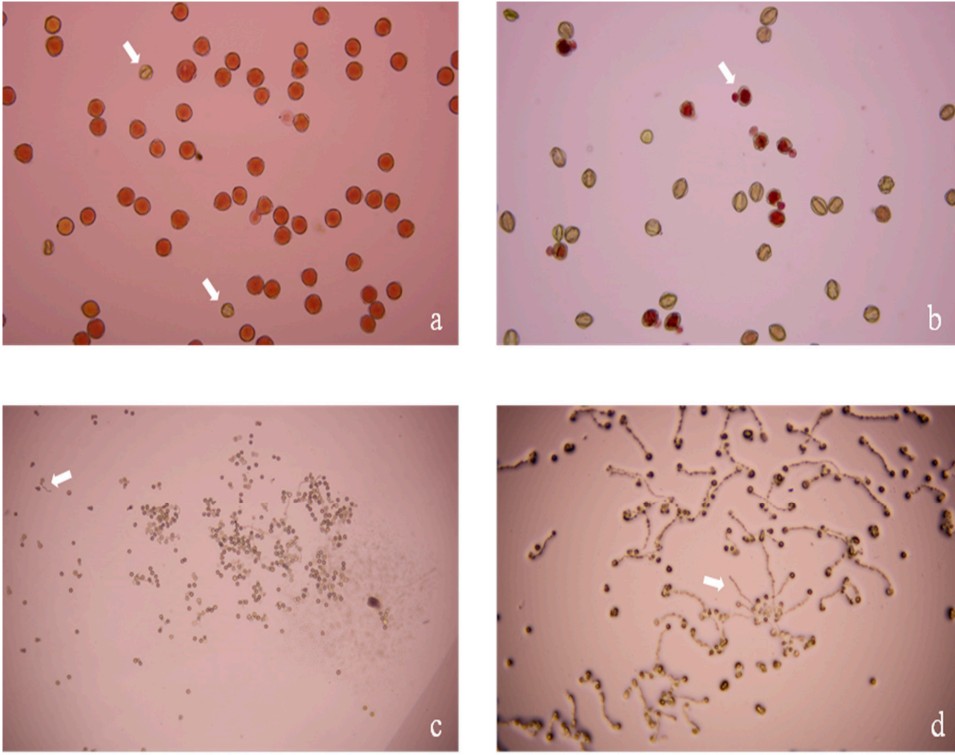

**Figure 2.** (**a**) Pollen grains stained with aceto-carmine (AC); (**b**) Pollen grains stained with 2, 3, 5-triphenyltetrazolium chloride (TTC). The arrow points to non-viable grains, (**c**,**d**) sprouted pollen grains on culture medium, directly on the plate. The arrow points to the pollen tube.

The germination medium (based on the [48] medium) contained 200 g sucrose, 0.1 g $H_3BO_3$, 0.3 g Ca $(NO_3)_2$, and 20 g of agar. Four-section, 90 mm plastic Petri dishes (Phoenix Biomedical SL, Murcia, Spain) were used for this purpose. Hydrated pollen was distributed over each of the four compartments containing solid medium [46]. To avoid dehydration and improve the surface area, 10 µL of the liquid culture medium (without agar) was added with a sterile pipette over each quadrant, which also favors

the start of germination [49]. After 24 h incubation in the dark at 25 °C [50], the Petri dishes were directly observed under an Olympus CX3 (×100 magnification) light microscope (Olympus Iberia S.A.U., Barcelona, Spain). To determine the germination rate, at least 100 pollen grains were counted in each compartment. The in vitro germination rate was determined by counting at least 100 pollen grains per preparation, considering as germinated the pollen grains with a pollen tube length equal or larger than the grain diameter [49,51,52].

### 2.4. Statistical Analysis

To determine the association degree between the number of flowers, number of berries, pollen viability, germination rate, pollen production per anther, and pollen production per vine, we applied a Spearman's correlation test and we obtained a correlation matrix. We applied this non-parametric statistical analysis since our data did not adjust to a normal distribution. The considered variables for the analysis were independently considered for each variety, vineyard, and year, building the data sets of: Godello Cenlle 2017, Godello Souto 2017, Godello Cenlle 2018, Godello Souto 2018, Mencía Doade 2017, Mencía Souto 2017, Mencía Doade 2018, and Mencía Souto 2018. We used the packages 'PerformanceAnalytics' 2.0.4 and 'corrplot' 0.84 of the R software 3.6.1 (R Core Team, R Foundation for Statistical Computing, Vienna, Austria) with editor RStudio 1.1.463 (RStudio Team, Boston, NY, USA) for statistical computing. Significance was calculated at 95% and 99% confidence level, * $p < 0.05$, and ** $p < 0.01$.

## 3. Results

### 3.1. Climatic Characterization

The annual average mean temperature was similar in the Ribeiro and Ribeira Sacra regions (Table 1), although it was slightly higher in the Ribeiro region in 2018 with 15.2 °C compared to 14.2 °C in Ribeira Sacra. The average maximum temperatures were markedly higher in the Ribeiro region in 2017 and 2018 with almost 3 °C difference. On the contrary, the average minimum temperature was higher in Ribeira Sacra in both years, with 7.9 °C in 2017 and 9.2 °C in 2018.

**Table 1.** Annual values of the main meteorological variables considered for both wine-growing regions, Ribeiro and Ribeira Sacra in 2017 and 2018.

| | | Ribeiro | | Ribeira Sacra | |
|---|---|---|---|---|---|
| | **Year** | **2017** | **2018** | **2017** | **2018** |
| **Annual Average** | Tmax (°C) | 23.9 | 23.7 | 21.2 | 21.2 |
| | Tmin (°C) | 6.4 | 8.2 | 7.9 | 9.2 |
| | Tmean (°C) | 13.9 | 15.2 | 13.6 | 14.2 |
| | Mean RH (%) | 75.8 | 75.9 | 74.3 | 77.5 |
| | Sunshine (hours) | 5.9 | 4.9 | 5.7 | 4.6 |
| **Annual Total** | Rainfall (L/m$^2$) | 814.5 | 775.2 | 693.9 | 804.4 |
| **Maximum** | Daily rainfall (L/m$^2$) | 100.8 | 39.4 | 64.9 | 39.2 |
| | Date | Dec. 10th | Feb. 28th | Dec. 10th | Mar. 11th |

Although both winemaking regions belong to the interior part of Galicia, the different disposition of these valleys influence the amplitude of temperature annual oscillation. The Rivas-Martínez [37] continentality Index (Ic) classified both Ribeiro and Ribeira Sacra regions in the same type of continentality, within the Oceanic type and Euoceanic subtype, but at different levels. The Ribeiro region was classified as Euoceanic-attenuated (category 2.2b) and the Ribeira Sacra as Euoceanic-pronounced (category 2.2a), which indicates a slightly higher continentality for the Ribeiro region (with a higher temperature amplitude, of 16.1 °C) in comparison to the Ribeira Sacra region (with 14.6 °C of amplitude).

For relative humidity and sunshine hours, we did not observe notable differences among the study years and plots. The highest daily rainfall record was observed in 2017 in the Ribeiro region, with 100.8 L/m$^2$ on 10 December, which coincided with the rainiest year, with a total sum of 814.5 L/m$^2$. The second rainiest year was 2018 in Ribeira Sacra with a sum of 804.4 L/m$^2$ (Table 1).

### 3.2. Production of Flowers and Berries: Fruitset and Coulure Indices

The number of flowers in control bunches was homogenous in the different vineyards during the study period for Godello variety, ranging from 136–198 flowers/bunch (Figure 3a). In the case of Mencía variety, important variations were observed between a minimum of 181 flowers/bunch in Souto during 2017 and a maximum of 318 flowers/bunch in Doade during 2018 (Figure 3c). The production of berries/bunch followed a similar trend to the production of flowers/bunch, with a narrow range for Godello values, ranging from 97–128 berries/bunch, and higher variability for Mencía variety that ranged between 149 berries/bunch (in Souto 2018) and 190 berries/bunch (in Doade 2018) (Figure 3a,c).

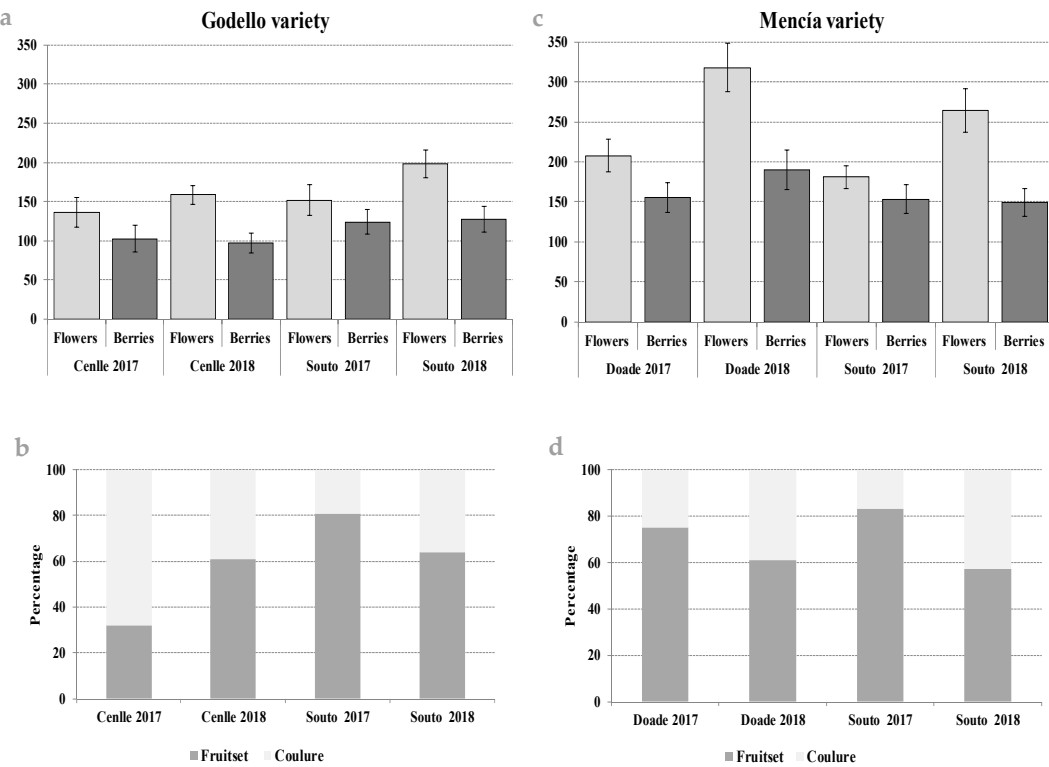

**Figure 3.** Average values of flowers and berries in sampled bunches for Godello (**a**) and Mencía (**c**). Fruitset %, and coulure % index for the Godello (**b**) and Mencía (**d**) varieties, in the studied vineyards during 2017 and 2018 (standard error of the mean on each column).

The mean production of flowers was higher in 2018 for both varieties and all study plots. However, the mean production of berries showed no important differences among study years, being slightly higher in 2017 for both varieties in all study plots, except for Godello in Souto and Mencía in Doade, with a higher berry production in 2018 than in 2017, with a difference of four berries in Souto and 34 berries in Doade (Figure 3a,c).

For further analysis of the relationship between flowers and berries in both grapevine varieties, we applied a lineal regression analysis between them for each study plot (Figure 4). We obtained significant regressions for all the considered combinations of variety-plot-year, with significance level of 99% ($p < 0.01$) for Godello in Cenlle 2017 (Figure 4a), Godello in Souto 2017 (Figure 4c), and Mencía in Souto 2017 (Figure 4g), and with significance level of 95% ($p < 0.05$) for the rest of combinations (Figure 4). Moreover, the three study plots with the highest significance level also showed the highest

$R^2$ adjusted coefficient of determination, higher than 0.8. This result showed that the number of flowers and berries kept balanced, with no important discordances on each analyzed cluster.

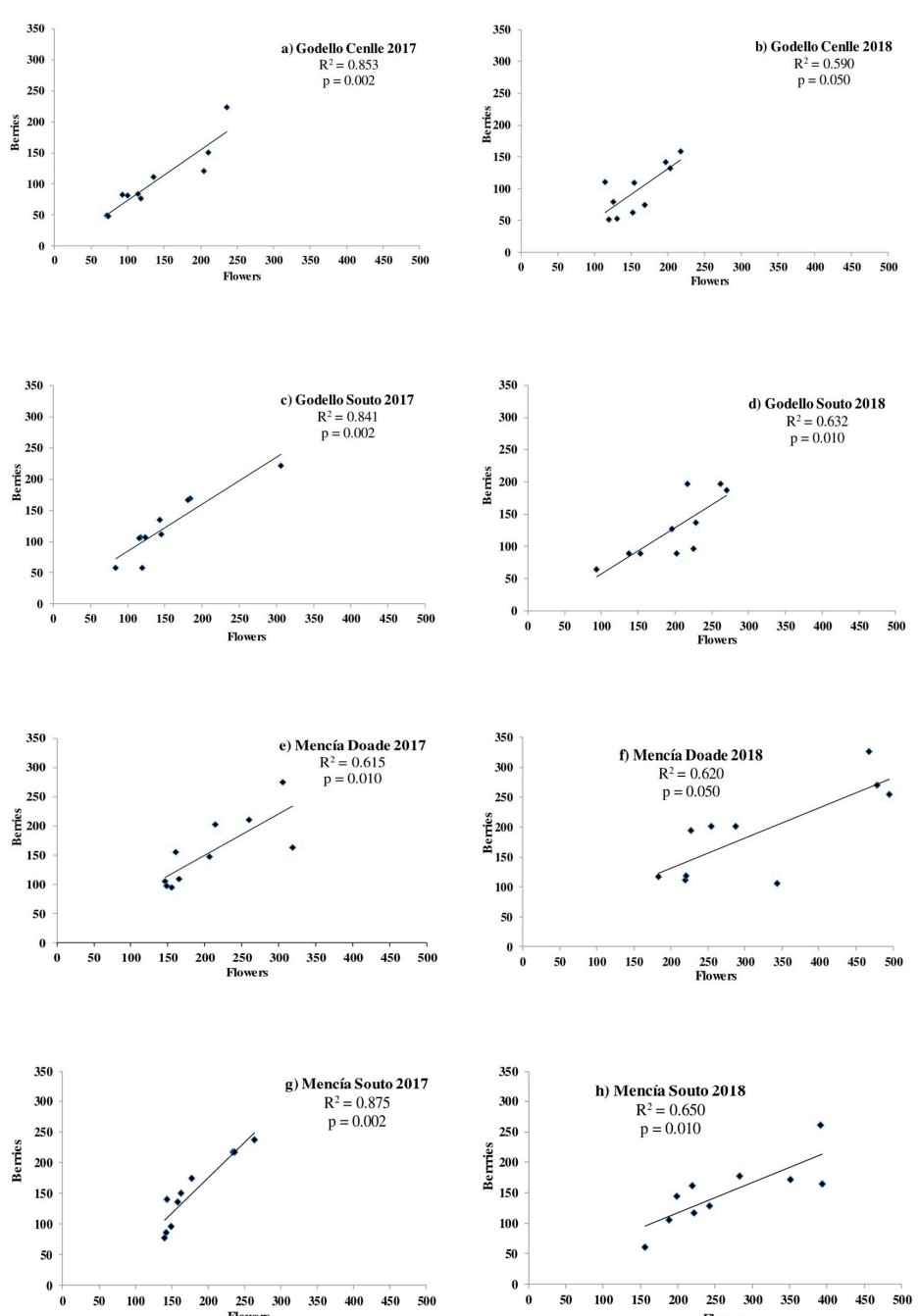

**Figure 4.** Linear regression analysis between flowers and berries in both grapevine varieties studied per plot, $R^2$ adjusted coefficient of determination, and significance level (p-level) in: (**a**) Godello Cenlle 2017 (Spearman's rank correlation = 0.9); (**b**) Godello Cenlle 2018 (Spearman's rank correlation = 0.63); (**c**) Godello Souto 2017 (Spearman's rank correlation = 0.9); (**d**) Godello Souto 2018 (Spearman's rank correlation = 0.85); (**e**) Mencía Doade 2017 (Spearman's rank correlation = 0.84); (**f**) Mencía Doade 2018 (Spearman's rank correlation = 0.68); (**g**) Mencía Souto 2017 (Spearman's rank correlation = 0.96); (**h**) Mencía Souto 2018 (Spearman's rank correlation = 0.81). The continuous lines represent the linear regression lines.

Comparatively with Figure 3, for both Godello in Souto 2017 and Mencía in Souto 2017, the high $R^2$ coefficient could reflect an efficient fruitset, as observable in Figure 3, with high fruitset rates for the

study plot and year (81% and 83%, respectively). In the case of Godello in Cenlle 2017, with a high $R^2$ coefficient, too, the fruitset rate in Figure 3 was low (32%), since this value represented the mean of the 10 analyzed vines on each plot, and the regression analysis showed the adjustment of each of the 10 individual data points to the regression line. Despite some of data being more dispersed, this effect was more pronounced in the mean fruitset value.

For the rest of combinations, the $R^2$ coefficients were lower, around 0.6 in all cases, which indicated a higher data dispersion and a slightly worse adjustment to the regression line.

The 50% of fruitset was far exceeded in all studied vineyards (Figure 3), except for Godello in Cenlle during 2017 (Figure 3b,d). This variety reached its maximum percentage of fruitset in the Souto vineyard in 2017, with 81.3% of fruitset, very close to the fruitset of Mencía in the same vineyard and year, with 83.1%.

### 3.3. Pollen Production, Viability, and Germination Ability

The mean pollen production per anther was clearly higher in 2018, independent of variety or vineyard location (Figure 5a). For the Godello variety, it varied between 1800–5540 pollen/anther, corresponding to Cenlle in 2017 and Souto during 2018, while for Mencía, it varied between 1250–5980 pollen/anther in Souto during 2017 and Doade in 2018.

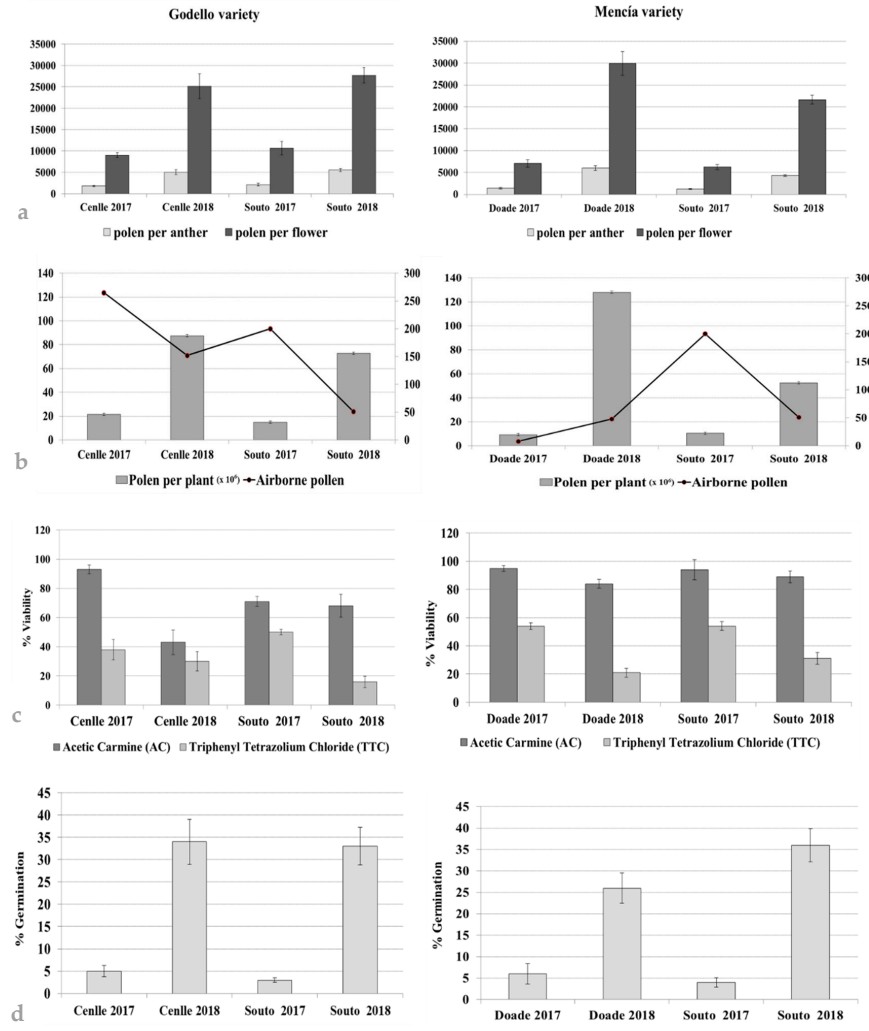

**Figure 5.** (**a**) Average values of pollen per anther and pollen per flower. (**b**) Average value of pollen per plant ($10^6$) (left axis) and airborne pollen (right axis). (**c**) Percentage of pollen viability by means of aceto-carmine (AC) and 2, 3, 5-triphenyl tetrazolium chloride (TTC). (**d**) Germination rates for Godello and Mencía varieties in the studied vineyards (standard error of the mean on each column).

The production of pollen per vine, obtained from the multiplication of the number of flowers by the pollen/flower of each variety, showed notable differences among study years and plots. For Godello, the pollen/vine production fluctuated between 14,917,500 pollen grains in Souto during 2017 and 87,578,050 pollen grains in Cenlle during 2018, while for Mencía the pollen/vine production varied between 9,238,840 pollen grains in Doade during 2017 and 128,061,700 pollen grains in Doade during 2018 (Figure 5b).

The airborne *Vitis* pollen concentration did not show a direct relation with the production of pollen per vine for any of the considered varieties (Figure 5b). The maximum airborne pollen value along the *Vitis* growth cycle was found in Cenlle during 2017 with 265 pollen grains, which coincided with one of the lowest values of pollen/vine production for the Godello variety. The same pattern was observed for the Mencía variety since the highest pollen/vine production in the Doade vineyard did not correspond with the airborne pollen recorded.

The obtained results of pollen viability by the AC method were greater than the obtained by the TTC methods in all cases, being, as well, higher in 2017 than 2018 (Figure 5c). For the Godello variety, the viability values varied between 43–93% by AC and 16–50% by TTC, with maximum values in Cenlle during 2017 by the AC method and Souto in 2017 by the TTC method. Regarding the Mencía variety, the pollen viability values ranged from 84 to 95% by AC and 21 to 54% by TTC, with more homogenous results along the study years. The maximum viability percentage obtained by both methods was found in Doade during 2017, although with low difference with respect to the obtained in Souto for the same year.

The percentage of pollen tube germination was much higher in 2018 for both grapevine varieties with values greater than 30%, while in 2017 they were lower than 10% (Figure 5d).

### 3.4. Grape Production and Its Relation with Pollen Production, Viability, Germination Rate, and Fruitset

According to the information provided by the wine-producing holdings where this study was carried out, the Godello harvest was markedly higher in the Cenlle vineyard in 2018 than the previous year, with 15,600 kg/ha, which represents more than double from the previous year yield. In the Souto vineyard, no important variations were observed, collecting around 2500 kg/ha every year. The Mencía yield was notably smaller than the Godello one, with maximum yield of 7950 kg/ha in Doade during 2017 and minimum of 3418 kg/ha in Souto during 2018 (Figure 6).

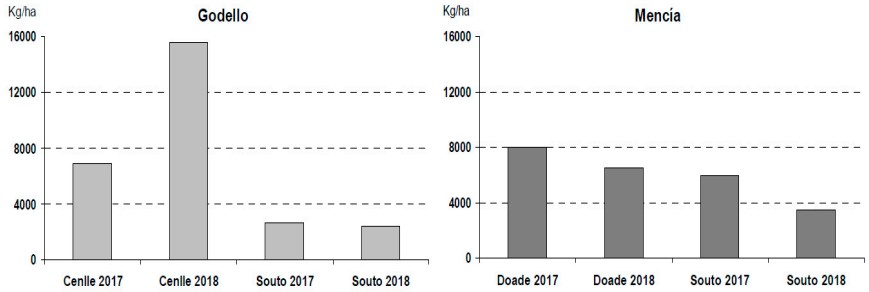

**Figure 6.** Production of grape (kg/ha) of Godello and Mencía varieties in each vineyard and studied year.

The excellent yield of Godello in 2018 with respect to the previous year can be associated with a higher production of pollen per plant, fruitset percentage, and germination rate. The best Mencía yield in the Ribeira Sacra vineyards, in Doade 2017, corresponded with a higher fruitset percentage and pollen viability (by the two considered methods of AC and TTC).

### 3.5. Statistical Analysis

As a result of the applied Spearman's correlation test, significant correlations were obtained between flower and berry production (flowers/berries) for all possible grapevine variety-vineyard-year combinations, with positive Spearman's rank coefficient above 0.636 and signification level up to 95% ($p < 0.05$) (Figure 7).

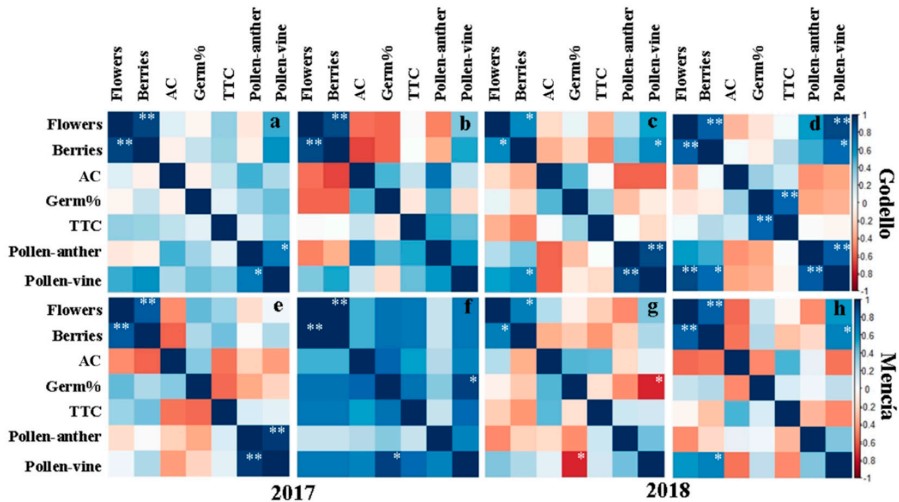

**Figure 7.** Spearman's rank correlation matrix coefficients between flower and berry number, pollen grains' viability by means of aceto-carmine (AC) and 2, 3, 5-triphenyltetrazolium chloride (TTC), germination rate, and pollen production per anther and vine. Squares a, b, c, and d correspond to Godello variety, and the squares e, f, g, and h to Mencía. The four left squares correspond to 2017 and the four right squares to 2018. Godello in: (**a**) Cenlle 2017, (**b**) Souto 2017, (**c**) Cenlle 2018, (**d**) Souto 2018; Mencía in: (**e**) Doade 2017, (**f**) Souto 2017, (**g**) Doade 2018, (**h**) Souto 2018. Coefficient values were expressed as colors (legend in the right of the graphs), and significance level was represented as * $p < 0.05$, ** $p < 0.01$.

The correlation between the pollen production per anther and vine (pollen anther/pollen vine) was highly significant for Godello in Cenlle and Souto, with the exception of Souto in 2017. For Mencía, this correlation was only significant in Doade during 2017 (Figure 7).

The relation between the production of pollen per vine and berries (pollen vine/berries), which reflects the vines' yield based on pollen production, was statistically significant for Godello variety during 2018, in both Cenlle and Souto and for Mencía variety in Souto during 2018. Furthermore, a strong correlation was found between pollen per vine and flowers (pollen vine/flowers) for Godello in Souto during 2018, which represents the reproductive performance of this variety.

The germination rate of grapevine pollen showed significant correlations with pollen viability by TTC method (Germ%/TTC) for Godello in Souto during 2018, and with pollen per vine (Germ%/pollen vine) for Mencía in Doade during 2018 and Souto in 2017 (Figure 7, Table 2).

**Table 2.** Spearman's rank correlation coefficients for different parameters of the studied varieties (*p* level: * < 0.05, ** < 0.01).

| | **GODELLO** | | | |
|---|---|---|---|---|
| | Cenlle | Cenlle | Souto | Souto |
| | 2017 | 2018 | 2017 | 2018 |
| Flowers/Berries | 0.903 ** | 0.636 * | 0.927 ** | 0.812 ** |
| Pollen Anther/Pollen Vine | 0.718 * | 0.894 ** | 0.580 | 0.827 ** |
| Pollen Vine/Berries | 0.600 | 0.636 * | 0.522 | 0.763 * |
| Pollen Vine/Flowers | 0.491 | 0.539 | 0.371 | 0.891 ** |
| Germ%/TTC | 0.104 | 0.317 | −0.279 | 0.794 ** |
| | **MENCÍA** | | | |
| | Doade | Doade | Souto | Souto |
| | 2017 | 2018 | 2017 | 2018 |
| Flowers/Berries | 0.842 ** | 0.681 * | 0.964 ** | 0.818 ** |
| Pollen Anther/Pollen Vine | 0.934 ** | 0.401 | 0.657 | 0.389 |
| Pollen Vine/Berries | 0.262 | 0.267 | 0.714 | 0.672 * |
| Germ%/Pollen Vine | −0.217 | −0.742 * | 0.912 * | 0.226 |

## 4. Discussion

Fruitset especially influences grapevine production, with a different degree of berry set self-regulation for each grapevine variety. May [53] stated that only 50% of flowers will form grapes. Bunches with high fruitset rate are more compact, therefore, more susceptible to fungal infections, while bunches with fruitset rates less than 30% will form laxer clusters with grapes of different sizes. Callejas et al. [54] denoted that generally, as the number of flowers increases, the fruitset percentage decreases. Our results did not reflect this statement since we found a positive statistical correlation between the production of flowers and berries for all considered spatio-temporal situations. Previous field studies carried out in Cenlle showed fruitset values for Godello variety around 79% [55], similar to the obtained values in the present work in 2018 of 61%, but very distanced from 32% obtained in 2017. This could be related with rainfall events that occurred on previous days or during flowering in 2017 of 125 L/m$^2$, which were much higher than rainfall for the same period in 2018 of 32.2 L/m$^2$. Rain can inhibit pollination and fertilization by diluting pollen grains from the stigma surface [53].

We considered the coulure index for the evaluation of the reproductive performance of the studied varieties to ensure an adequate fertilization on each study plot and year. The highest value of 68% corresponded to Godello (Cenlle 2017), while the lowest coulure value of 17% was registered for Mencía (Souto 2017). Despite these results, previous studies pointed out high fruitset rates for Godello variety, higher than 90% [23], which means that the highest coulure value found in the present study, of 68% for the Godello variety in Cenlle 2017, could be due to environmental conditions, such as meteorological or phytopathological conditions, which could affect the fertilization process and fruitset. In general, the mean coulure value for each variety was also higher for Godello, with 40.5%, than for Mencía, with 31%. In addition to possible differences due to variety characteristics, variations on the coulure index between the considered regions, Ribeiro and Ribeira Sacra, could be related with the temperature regimes of these two climate subtypes. The applied continentality index classified both regions as Oceanic, but the Ribeiro region showed more continentality with higher annual temperature amplitude of 2 °C above the Ribeira Sacra value. The Ribeiro average maximum temperatures were markedly higher with almost 3 °C difference, in contrast to the colder average minimum temperatures of this Euoceanic-attenuated climatic region. Pagay and Collins [56] pointed out the negative effect of the increase in the average growing season temperature on crop yield, with a reduction of 10% in fruitset with heat periods at flowering. They found a significant increase of the coulure index value at flowering with a 2–4 °C temperature increase, indicating a higher level of flower abscission that was not fertilized under elevated temperatures.

Our results showed a direct relation between the production of pollen per vine and the number of flowers by the statistical Spearman's correlation test. The positive significant correlation of pollen vine/flowers found for Godello in Souto 2018 could reflect the plant reproductive performance expressed as the relation between the number of flowers and the total pollen produced by the vine, which also depends on other factors such as the production of pollen per anther, the production of bunches per plant, or meteorological conditions. In addition, we also detected that the number of flowers produced by the two studied varieties were proportional to the coulure index, despite no significant correlation being found.

Pollen production per flower varies depending on the grapevine variety. The varieties Siyad genere and Razaki grown in Isparta, Turkey, produce between 3000–9000 pollen grains/flower [18]. The Treixadura variety produces around 9905 pollen/flower [55], Brancellao 11,060 pollen/flower, and Merenzao 4185 pollen/flower [57], while Tempranillo and Torrontés exceed the 20,000 pollen grains per flower [58]. These values are close to the mean pollen/flower values obtained for the analyzed varieties in the present study, of 18,114 pollen/flower for Godello and 16,221 pollen/flower for Mencía.

Considering the pollen production per vine, the Godello plants reached maximum in Cenlle 2018 with more than 87 million pollen grains/vine, which coincided with the highest grape production for this variety, of 15,600 kg/ha. Fernández-González et al. [55] indicated lower values for this variety, around 35 million pollen grains per vine. In the case of Mencía, the highest record of pollen/vine

was obtained in Doade during 2018 with 128 million pollen grains. However, this maximum value of pollen production per vine was not reflected on grape production, as it remained second in grape production volume after the Doade during 2017 production. These differences could be associated with the number of final bunches on the plant, which is determined by the genetic trait of each variety, but it is affected by the green pruning system, with the winegrower thinning them out to a greater or lesser extent [59]. The espalier training system used in the Cenlle vineyard, where the maximum grape harvest was obtained, increases the leaf area and, consequently, the number of bunches per plant, while the Souto goblet-trained vines facilitate bunches' spacing, which decreases its number. Our statistical results showed this concordance of the production of pollen per vine and the grape production as we found positive significant correlations between pollen per vine and berries for Godello in Cenlle during 2018 and Souto in 2018, and Mencía in Souto along 2018.

Environmental and meteorological conditions have a marked influence on pollen release and dispersion processes, strongly linked to anther dehiscence [60,61]. These factors are determinant for the final grape harvest, previously defined by the endogenous pollen generation, influencing the effectiveness of the fertilization. We observed important discordances between airborne pollen concentrations and pollen produced by vines, which can be due to the influence of weather conditions. The dissimilar behavior of airborne pollen and the production of endogenous pollen per plant was observed for both varieties in two directions: (1) High pollen production by the plant and low airborne pollen concentrations or (2) low pollen production and high airborne pollen concentrations. The first situation was detected for Godello in Souto during 2018 and Mencía in Doade during 2018, with airborne pollen concentrations of 51 and 48 pollen/m$^3$, respectively. These low pollen concentrations, in comparison with the other values, coincided with elevated pollen productions per vine, being, in the case of Doade in 2018 the maximum production among the study cases, of $128 \times 10^6$ pollen grains per plant. This dissimilar trend detected could be related with self-pollination processes, an important floral mechanism that often occurs before capfall although cross-pollination also takes place, which improves seed set in berries and increases fertilization effectiveness [45]. This mechanism was previously reported for the Godello variety in the Ribeiro region vineyards [55]. The second inverse situation, of low pollen production per plant and high airborne pollen concentrations was observed for Godello in Cenlle 2017, Souto 2017, and for Mencía in Souto 2017. These inverse values could reflect the influence of environmental conditions, since favorable conditions for pollen release and dispersal, such as temperatures around 20 °C with dry weather and slight wind, can enhance the presence of pollen in the atmosphere. These meteorological conditions are considered as optimal for grapevine pollination [62,63]. Humidity conditions are also relevant for a correct dehiscence and pollination, since anthers have to dry out to this process but too dry and windy conditions can dry the stigma surface hindering pollen from sticking to it [45]. Rainy conditions affect the calyptra opening, remaining attached to the pistil, causing difficult pollination [64]. Periods with continuous rain and temperature below 15 °C lead to pollen washing from the atmosphere and pollen grains' agglomeration [29–65]. In the case of Mencía, in Souto during 2017 we recorded the highest airborne pollen concentration for this variety of 200 pollen/m$^3$, coinciding with one of the lowest values of pollen production per vine, of $10.6 \times 10^6$ pollen grains. This could explain why grape production was not markedly lower in this year, of 5916 kg/ha, despite the low pollen production per plant. For Godello in Cenlle during 2017, we registered the highest airborne pollen concentration of 265 pollen/m$^3$, low pollen production per vine ($22 \times 10^6$ pollen grains), and a production of 6898 kg/ha. In Souto during 2017 for the same variety, the airborne pollen was slightly lower with 200 pollen/m$^3$, which was accompanied by a lower pollen production per plant ($15 \times 10^6$ pollen grains) and a lower grape production, of 2640 kg/ha. These values showed the importance of the dispersion of airborne pollen on the fertilization process.

Meteorological conditions near to flowering phenological stage and pollen quality, defined as viability, germination ability, and fertility, are fundamental factors for fruit development [66]. Pollen germination ability is related to variety, nutritional conditions, and environmental factors, with a

great variation in optimum germination conditions among plant species and varieties [18]. In the present study, the pollen viability values were quite stable for each variety, especially for Mencía, with a standard deviation (SD) of 5.07 and coefficient of variation (CV) of 0.06 for AC method, and SD of 16.67 and CV of 0.42 for TTC method. The variability in pollen viability values increased for Godello variety, but it was still stable for each method, with a CV lower than 1 (0.30 for AC and 0.43 for TTC) (Figure 5c). Regarding the germination rates, the germination values were markedly lower in 2017, with important differences with respect to 2018 values (Figure 5d) and, considering that the study plots and vines were the same for both years, the environmental conditions in 2017 could have negatively affected pollen germination. Temperatures from 27 to 28 °C [67,68] and up to 32 °C [69] are considered as optimal temperature ranges for ovule germination and growth of the pollen tube. Thermal stress caused by heat or drought events produces several abnormalities in the reproductive organs' structure and functioning, affecting processes such as pollen grain development, ovule fertility, or berry growth in many *Vitis vinifera* varieties [70]. Exposure to near to 40 °C temperatures during 4 h can lead to a 50% reduction in germination capacity [28]. On the contrary, exposure to low temperatures near flowering also negatively affects pollen germination, pollen tube growth, and ovule development [45]. Ebadi et al. [71] pointed out that exposure to a temperature regime of 12 °C/9 °C day/night near to flowering caused a reduction in the number of pollen tubes developed in the lower ovary, and the ovules exposed to lower temperatures tended to be smaller and less developed than the control vines, kept at 25 °C/20 °C until flowering. Other studies indicated notable differences on germination rates among grapevine varieties, such as Tempranillo with 8.7%, Torrontés Riojano with 24.7% [58], Cabernet Sauvignon with 7–11.7%, or Loureiro with 46.7–60% [72]. Besides different extrinsic factors, such as climatology or grapevine phytosanitary conditions, several intrinsic factors can affect pollen germination as metabolic blocks, suppressor's presence, stigma pH, or pollen anomalies [73].

From the methodological point of view, we found considerable differences among the pollen viability values depending on the method used, with higher values for the AC method than the TTC, which agreed with the results pointed out by other authors. Castiñeiras et al. [46] applied different staining methods to study the pollen viability of *Fraxinus excelsior* and *Fraxinus angustifolia* (Oleaceae family), including the ACG (aceto-carmine glycerol) and TTC stains, and they found important differences for both methods applied to the same plant material. They found markedly lower pollen viability values using TTC staining, with a mean value of 22.4%, than those obtained by the ACG staining, with a mean pollen viability of 95.9%. Our results were similar to those obtained by Castiñeiras et al. [46]. For both Godello and Mencía varieties, pollen viability was higher with the AC method than the TTC method in all study plots and years, with a mean pollen viability for Godello of 68.75% by AC and 33.5% by TTC, and a mean pollen viability for Mencía of 90.5% by AC and 40% by TTC. The obtained results were also similar to those found by Naab et al. [58] for other grapevine varieties, such as Tempranillo (65.6%), Ugni Blanc (53.8%), and Torrontés Riojano (56.9%).

## 5. Conclusions

The studied variables to describe the reproductive performance of the grapevine Godello and Mencía varieties were useful to analyze the pollen effectiveness of each variety, relating these plant reproduction variables with the final grape production. The 50% of fruitset was far exceeded in all studied vineyards, except for Godello in Cenlle during 2017. In this vineyard and season, the 68% coulure value was reached by the Godello variety, which could have been due to environmental conditions (such as meteorological or phytopathological conditions) since this variety usually express higher fruitset rates. In comparison with the studied Mencía vines, the mean coulure value was higher for Godello (40.5%) than for Mencía (31%). In addition to possible differences due to variety characteristics, differences on the coulure index between the considered regions, Ribeiro and Ribeira Sacra, could be related with temperature regimes of these two climate subtypes since there is a higher annual temperature amplitude in the Ribeiro region than the Ribeira Sacra region.

Analyzing comparatively the endogenous pollen generation and airborne pollen levels, we observed important discordances between them, which could have been due to the influence of weather conditions. The detected low airborne pollen concentrations coincided with a high pollen production per plant in 2018 for Godello in Souto and Mencía in Doade. This could be related with self-pollination processes, an important floral mechanism that often occurs before capfall. On the other hand, we detected high airborne pollen concentrations that coincided with relatively low pollen production per plant for Godello in Cenlle 2017 and Souto 2017 and for Mencía in Souto 2017. In this case, these values could reflect the influence of environmental conditions, since favorable conditions for pollen release and dispersal during flowering, such as temperatures around 20 °C with dry weather and slight wind, can enhance the presence of pollen in the atmosphere.

We found important differences on pollen viability depending on the applied method, with higher values for the AC method than the TTC for both varieties and study plots. The higher values were observed for Mencía variety than Godello, which could be due to variety characteristics. Regarding germination rates, we observed a marked reduction in 2017 with respect to 2018, in all study sites and for both varieties. The environmental conditions could negatively affect pollen germination in 2017.

Besides the studied pollen and production variables, other factors have important influence on wine production, such as the meteorological conditions along the vegetative cycle, edaphic parameters (such as effective soil depth or soil water reserve), the phytopathological state, or agronomic practices.

**Author Contributions:** Investigation and study design, J.A.C., F.J.R.-R., and M.J.A.; methodology, M.F.-G. and E.G.-F.; analysis and interpretation of data, R.A.V.-R.; manuscript preparation and final review, J.A.C., F.J.R.-R., M.J.A., M.F.-G., E.G.-F., and R.A.V.-R. All authors have read and agreed to the published version of the manuscript.

**Funding:** This research was funded by the Xunta de Galicia (Consellería de Educación, Universidade e Formación Profesional) through the recognition as Grupo de Referencia Competitivo de Investigación (GRC GI-1809 BIOAPLIC "Biodiversidad y Botánica Aplicada", ED431C 2019/07), the Agrupación Estratégica de Investigación BioReDes (ED431E 2018/09) and the BV1 Reference Competitive Research Groups ED431C 2017/62 (Xunta de Galicia, Spain). This work was partially funded by Xunta de Galicia CITACA "Cluster de Investigación y Transferencia Agroalimentaria de Campus del Agua" Strategic Partnership (Reference: ED431E 2018/07) and the AGL2014-60412-R Economy and Competence Ministry of Spain Government project. Fernández-González M. was supported by FCT "Fundação para a Ciência e a Tecnologia" (SFRH/BPD/125686/2016) through the HCOP-Human Capital Operational Program, financed by "Fundo Social Europeu" and "Fundos Nacionais do MCTES (Ministério da Ciência, Tecnologia e Ensino Superior). González-Fernández E. was supported by the Ministry of Sciences, Innovation and Universities (FPU "Ayudas para la Formación de Profesorado Universitario" grant FPU15/03343).

**Acknowledgments:** The authors thank the Xunta de Galicia (Consellería de Educación, Universidade e Formación Profesional) for the financial support and the Agrupación Estratégica de Investigación BioReDes (ED431E 2018/09).

**Conflicts of Interest:** The authors state that there is no conflict of interest regarding the publication of this article.

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
