# Peer review of "Potential Fertilization Capacity of Two Grapevine Varieties: Effects on Agricultural Production in Designation of Origin Areas in the Northwestern Iberian Peninsula"

_agronomy, doi:10.3390/agronomy10070961_

Round 1

Reviewer 1 Report

The article is a report on the fertilization capacity of two grapevine varieties. While the report is of some interest for the readers, additional correction by a scientific English translator is recommended before resubmission. All in all, English should be revised throughout the manuscript. Many sentences should be rewritten and English grammar should be checked.

Also throughout the manuscript, there are some inconsistencies. For example, why is phenological stage 'fruitset' written as such while "Coulure" is written with a capital 'C'? And throughout the manuscript, in some cases these words are written with capital first letters and in other cases, without capital first letters. 

Throughout the manuscript, Vitis vinifera and the subspecies must be written in italics.

However, while it is a manuscript related to viticulture, there is no specific information related to the varieties under study. No morphological characters of the flower are mentioned (for example, how many stamens there are in the flower), which is very important to mention especially in such studies.

Discussions needs to be revised. Lines 283-284 do not make sense. 

In lines 318-119 Razaki is mentioned as a Turkish variety. Please revise and double check. Razaki is considered a variety of eastern origin, but not Turkish.

In lines 330-333 it is written that the number of bunches training system depends on the training system which is not valid. It is the pruning system that mainly affects the number of bunches that will be produced by the vine. Obviously, the factor variety plays the most important role since the number of inflorescences that will be created within the latent buds is a genetic trait of each variety.

In lines 377-378 the authors assume that the presence of acolporated pollen form is a cause for low germination rates. Couldn't this have been checked with an electronic microscope?

Reviewer 2 Report

The manuscript presents interesting data on reproductive traits such as number of flowers, number of berries, fruitset, pollen production, viability and ability of germination. In general it is well-written and understandable.

For some variables it shows very dispersed data, in some cases contradictory, what authors called ‘dissimilar behavior’, that probably reflects the high dependency of some of these variables on environmental conditions. The first comment is that this environmental influence on the results is not highlighted enough on the Discussion, although it is in the Conclusions. And some farfetched explanations for contradictory results may simply be deleted, like thatin L356 and following.

Title: ‘Designation Origin Areas’ is this correct? Shouldn’t it be Designation of Origin Areas or Origin Denomination areas?

Abstract: L25-26: where the highest Coulure value was reached. This indicates a higher reproductive effort for this variety that would need to develop more flowers and pollen in anthers to ensure fertilization. What do we know about the ’needs’ of the variety? It is the grape grower needs, not the variety. So, do not express it in that way. The same in Discussion L298-299.

Introduction

Introduction is poor. There are not many works related to reproductive traits in grape, and authors do not introduce (nor discuss) some of the most relevant ones such as:

Collins C, Dry PR. Response of fruitset and other yield components to shoot topping and 2-chlorethyltrimethyl-ammonium chloride application. Aust J Grape Wine Res. 2009;15(3):256-67.

Dry PR, Longbottom ML, McLoughlin S, Johnson TE, Collins C. Classification of reproductive performance of ten winegrape varieties. Aust J Grape Wine Res. 2010;16:47-55.

Ibáñez J, Baroja E, Grimplet J, Ibáñez S. Cultivated Grapevine Displays a Great Diversity for Reproductive Performance Variables. Crop Breed Genet Genom. 2020;2(1):e200003.

Tello J, Montemayor MI, Forneck A, Ibáñez J. A new image-based tool for the high throughput phenotyping of pollen viability: evaluation of inter- and intra-cultivar diversity in grapevine. Plant Methods. 2018;14(1).

Pereira HS, Delgado M, Avo AP, Barao A, Serrano I, Viegas W. Pollen grain development is highly sensitive to temperature stress in Vitis vinifera. Aust J Grape Wine Res. 2014;20(3):474-84.

Instead, authors cite other works that are less relevant here, for instance the statements in L46-52: ‘The genetic characteristics […] to ensure a successful vintage [14-15-16]’ are of lesser relevancy for this work.

L55: ‘Some of them are self-incompatible’ Which grapevine varieties are self-incompatible?? Include citation or delete.

Material and methods

As it is calculated, Coulure is exactly (mathematically) 1-fruitset (or 100-fruitset, in %). It has no sense to represent both values in Figure 3b,d (in each case fruitset and coulure values sum up 100%), or to present or discuss them as if they were independent, like in the Discussion L289-294 for fruitset, mentioning 32% fruitset of Godello in 2017 in Cenlle, and then in L295-301 for coulure, mentioning 68% (=100-32) coulure for Godello in 2017 in Cenlle. In fact, Coulure refers to a different concept, according to Collins and Dry 2009, and Dry et al. 2010, where seedless berries and live green ovaries (LGO) are considered in addition to normal, seeded berries, which I assume are the ones counted in this work. If none of the clusters showed any seedless berries or LGO, then coulure is like it has been calculated in this work, but in that case fruitset and coulure are complementary values that can be treated (discussed) jointly.

Results

Figure 3: prepare it as Figure 5: Godello at the left and Mencia at the right. In a: flowers and berries (bars closer); in b, fruitset (if you want to include coulure, then use single bars of 100% with the corresponding fraction colored for fruitset and not colored for coulure or vice versa)

The differences found between the two staining methods used to estimate pollen viability are large. Is it normal? What have found other authors?

Inverse relationship of pollen viability with germination rate is weird; in fact, the only significant correlation, is positive (Godello-Souto-2018).

Figure 7. Improve the figure by adding self-explanation labels in the proper figure. The legend is very confusing. Besides, is the legend alright?: 2017 values are at the right squares and 2018 are at the left squares? In the rest of figures, it's the other way around.

Discussion

L341-366 Airborne pollen is not expected to be important in cultivated grapevine: ‘In contrast to their wild relatives and the few cultivars with female flowers, cultivated grapevines are typically self-pollinated, whereby pollen originates from the flower’s own anthers (Mullins et al., 1992; Sartorius, 1926)’. In Keller 2015, The Science of grapevines, pg 88. All this discussion paragraph is focused on the importance of the amount of pollen production on fruitset and yield. In fact, pollen is produced in a very large excess in every flower, which, on the contrary, produce only 4 ovules. Unless pollen production, and/or its viability and/or its ability to germinate are very seriously affected, pollen should not be the limiting factor in the fruitset process. Pollen studies are easier than ovule studies and are useful because sometimes they can provide evidences about possible troubles in the female gametogenesis. Pollen you studied seems ok, so I would not expect any impact on fruitset and yield, other than those caused by weather conditions, which are probably the main cause of the differences between airborne counts and pollen production. You should consider this in your discussion

Conclusions

What do you mean by ‘reproductive effort’??

Focus on the results, the differences between varieties, between places and between seasons.

 Do not mention ‘acolporated pollen forms’ because you have not seen them.

Minor changes

L45: change ‘are important factors to considered them both independently as’ by ‘are important factors to consider both independently as'  

L145-146 repeats what it is said in L140-141

L149-150 ‘we applied a Spearman´s correlation test calculate a correlation matrix’; amend this sentence structure

L306 ‘crop quality and yield’, delete quality, you did not study it

Figure 5 legend. Airborne pollen is not mentioned. Add what axis (left or right) represents what (pollen per plant ∙10-6 or airborne pollen) in b.

Reviewer 3 Report

In general, the paper is well organized and it has a good structure. Scientific problem has posed and it finds an answer in conclusions. Still, some minor revision is needed.

Some sentences need English language revison in lines: 138-140; 184; 283-284; 394.

Line 178 - Table 1. Seem to have an extra line which is empty or is there something missing?

Lines 186-187. The authors claim "The production of flowers and berries was higher in 2018 for both varieties (Figure 3a, c)." - when looking the figures, it seems to have it higher only for flowers, but not for berries in 2018. 

Line 800 - Figure 4. This figure is mentioned in the text only once (line 188) and the results are not explained in the results part at all.

Lines 2013-218. The paragraph needs reference to the adequate figure.

In the discussion part, the authors refer to an abbreviation - Coloure Index (CI) - that is never used as an abbreviation. I suggest to remove the abbreviation.

Lines 372 and 376 - the Latin names should be written similarly thoughout the manuscript.

Lines 420-426. Why are the financial support and partial funding indicated in acknowledgements, but not under funding (line 420)?

Round 2

Reviewer 1 Report

The authors took into consideration the reviewers' comments and suggestions, therefore the manuscript has been significanlty improved.